# Trends in Mortality Rates of Corpus Uteri and Ovarian Cancer in Lithuania, 1987–2016

**DOI:** 10.3390/medicina56070347

**Published:** 2020-07-13

**Authors:** Rūta Everatt, Birutė Intaitė

**Affiliations:** 1Laboratory of Epidemiology, National Cancer Institute, Baublio 3B, LT-08406 Vilnius, Lithuania; 2Department of Gynaecologic Oncology, National Cancer Institute, Santariškių 1, LT-08660 Vilnius, Lithuania; b.intaite@gmail.com

**Keywords:** corpus uteri cancer, ovarian cancer, average annual percentage change, mortality, trends, Lithuania

## Abstract

*Background*: The corpus uteri and ovarian cancers burden in Lithuania has remained high. The aim of this study was to investigate time trends in mortality rates of corpus uteri and ovarian cancer in Lithuania across age groups and time periods over a 30-year time span. *Materials and Methods*: Data on numbers of deaths from corpus uteri cancer during the period 1987–2016 and ovarian cancer during the period 1993–2016 were obtained from the WHO mortality database. Trends in age-standardized mortality rates (ASR, world standard), and age-specific rates were analyzed by calculating annual percentage change using Joinpoint regression. In addition, age–period–cohort analysis was performed for each cancer type. *Results*: Mortality from corpus uteri cancer decreased by −1.2% (95% CI: −1.8; −0.7) annually from 1987 to 2016. Decrease was most pronounced in youngest age group of 40–49 years; annual percentage change was −2.4 (95% CI: −4.0; −0.9). Mortality rates for ovarian cancers decreased by −1.2% (95% CI: −1.6; −0.8) annually from 1993 to 2016. Corpus uteri and ovarian cancer ASRs in 2016 were 3.5/100,000 and 7.4/100,000, respectively. The age–period–cohort analysis suggests that temporal trends in corpus uteri cancer mortality rates could be attributed to period and cohort effects. *Conclusion*: A reduction in mortality rate was observed for corpus uteri and ovarian cancer over the entire study period. Similar decreasing pattern for corpus uteri and ovarian cancer mortality indicate effect of shared factors.

## 1. Introduction

The incidence and mortality rates of corpus uteri and ovarian cancers vary widely among European countries [1,2]. In most developed countries, corpus uteri and ovarian cancer incidence and mortality have gradually declined since the 1990s [2]. However, relatively high rates are estimated in Central and Eastern Europe. Despite a decrease in rates observed during 1981–2008 [1,3], Lithuania ranks 3rd in Europe with respect to the corpus uteri cancer incidence rate (age standardized incidence rate = 24.0/100,000) and 6th with respect to mortality rate (age standardized mortality rate (ASMR) = 3.7/100,000) [1]. Based on GLOBOCAN 2018 estimates, Lithuania ranks 3rd in Europe according to ovarian cancer mortality (ASMR = 7.8/100,000) [1]. Corpus uteri cancer is 8th and ovarian cancer the 4th most common cause of cancer death among women in Lithuania (5% and 8% of all cancer deaths, respectively) [1]. It is unclear, what the recent pattern of corpus uteri and ovarian cancer mortality trends is and how age-specific mortality trends have contributed towards overall trends in Lithuania.

The aim of this study was to evaluate trends of corpus uteri and ovarian cancer mortality from 1987 to 2016 in Lithuanian population across age groups and time periods in the context of changes in the cancer treatment and prevalence of risk factors.

## 2. Materials and Methods

### 2.1. Data Source

We obtained data from the World Health Organization (WHO) mortality database online (the number of corpus uteri and ovarian cancer deaths and the female population size by each calendar year in 5-year age groups) [4]. Ovarian cancer mortality rates during 1987–1992 were unavailable from WHO database, thus 1993–2016 years were included in the analyses.

The following cancers were included: Cancer of corpus uteri (including the uterus not otherwise specified, NOS) and cancer of ovary. The cancer codes were used according to International Classification of Diseases (ICD) revisions 9th (1985–1997) or 10th (1998–2016), as described in Table 1. One ICD code B122 was used for both cancers of corpus uteri and uterus NOS in the 9th edition. In the ICD-10 edition C54 (corpus uteri cancer) and C55 (uterus NOS cancer) were used. Cancer of ovary was coded B123 (ICD-9) or C56 (ICD-10).

### 2.2. Analytic Methods

We used Joinpoint regression to analyze trends in age-standardized and age-specific cancer mortality rates by cancer type. We calculated annual age-standardized rates per 100,000 for each year using the direct method and the world standard population as reference [5]. Analysis was also carried out by age groups: 40–49, 50–59, 60–69, 70–79, and 80+. Using Joinpoint analysis we aimed to identify years at which point significant changes in trend occurred and estimate average annual percentage change (APC) for each trend segment identified by the model. A maximum number of three Joinpoints was allowed. *p*-values of < 0.05 were considered as statistically significant. Joinpoint trend analysis software, Version 4.5.0.1 (2017) was used [6].

With the aim of a more detailed analysis, to assess the effect of age, death period, and birth cohort on time trends, we performed an age–period–cohort analysis using the Web tool (http://analysistools.nci.nih.gov/apc/). For this purpose, data were grouped by 5-year age and period intervals, excluding women aged <20 years and ≥80 years. Ovarian cancer mortality rates during 1997–2016 years were included in the age–period–cohort analysis. From the Web tool we obtained: longitudinal age-specific rates (i.e., fitted age-specific rates in reference cohort adjusted for period deviations); period relative risk (adjusted for age and non-linear cohort effects in each calendar period versus the reference period) and cohort relative risk (adjusted for age and non-linear period effects in each given cohort versus the reference cohort). Additional details of the Web tool are described elsewhere [7].

We displayed the longitudinal age-specific rates, period and birth cohort effects graphically. In all age–period–cohort analyses, the reference group was the central calendar period and central birth cohort. We also obtained an estimate of the net drift, i.e., analogue of the estimated annual percentage change (APC) in the age-standardized mortality rate. Local drifts provide a model-based estimated APC value for each age group [7].

## 3. Results

A total of 4405 deaths from corpus uteri cancer were reported in Lithuania from 1987 to 2016. There were 6527 deaths from ovarian cancer during 1993–2016 (Table 2).

### 3.1. Age Standardized Mortality

Figure 1 shows ASMRs for corpus uteri and ovarian cancer in Lithuania. Mortality rates for corpus uteri cancer decreased by −1.2% (95% CI: −1.8; −0.7) annually, and for ovarian cancers by −1.2% (95% CI: −1.6; −0.8) annually throughout the study period. The ASMRs of corpus uteri and ovarian cancer in 2016 were: 3.5 and 7.5 per 100,000 women, respectively (Table 2).

### 3.2. Age-Specific Trends

Analysis of cancer mortality trends by age group showed that mortality rates of the corpus uteri and ovarian cancers steadily declined in all age groups except the oldest (70+), no Joinpoints were identified (Table 2, Figure 2). Women aged 40–49 years showed steeper decreases than older age groups. The corpus uteri cancer mortality rates were 3–4 times higher among women aged 60 years or older than in those aged less than 60 years. We also observed ovarian cancer mortality 3–6 times as high among women aged ≥50 years compared to younger women.

### 3.3. Age–Period–Cohort Analysis

Estimated age, period, and cohort effects of corpus uteri and ovarian cancer mortality are presented in Figure 3. The longitudinal age curve displays a steady rise in the risk of corpus uteri and ovarian cancer death in every successive age group (Figure 3a). The age–period–cohort analysis of corpus uteri cancer mortality showed significant cohort and period effects. The risk of dying due to corpus uteri cancer constantly declined with each subsequent birth cohort for women born after 1922 (Figure 3b). The period effect decreased until 2007–2011, then remained stable. Compared to reference period 1997–2001, mortality risk was significantly reduced in 2007–2011 and 2012–2016 with estimated relative risk of 0.73 (95% CI 0.57; 0.93) and 0.73 (95% CI 0,60; 0.88), respectively.

Our results show constantly decreasing probability of dying due to ovarian cancer for women born between 1922 and 1987 and an upward trend in women born after 1987s (Figure 3c). Sharply declining period effect was seen in ovarian cancer mortality.

Wald tests showed statistically significant cohort and period effects for corpus uteri cancer mortality (*p* < 0.05). Net drift shows that the overall change in corpus uteri cancer mortality in women during 1987–2016 was −1.48% (95% CI −2.14; −0.82) per year, *p* < 0.05. Local drifts were significant (*p* < 0.001), they were decreasing among women below 70–74 years and increasing in older women. Ovarian cancer mortality displayed a statistically significantly decreasing period effect (*p* < 0.05), but not cohort effect (*p* = 0.10). Net drift for ovarian cancer was −1.80% (95% −3.11; −0.48); *p* = 0.008. Local drifts were not statistically significant for ovarian cancer (*p* = 0.70).

## 4. Discussion

The main result of this study is the similarity of the corpus uteri cancer and ovarian cancer mortality trends, declining by 1.2% annually during the whole 30-years period. Similar pattern of birth cohort effects for corpus uteri and ovarian cancers suggests shared risk factors and decrease in their exposure. Similarity in period effects implies that factors such as the improved diagnosis and treatment could have played a role in the observed downward trends in ovarian and corpus uteri cancer mortality.

Previous findings from studies in Europe indicate, that after a decrease in corpus uteri cancer mortality, trends either continued to decrease, became stable, or increased in recent years [2,8,9]. Uterine cancer death rates were stable during entire 1995–2017 period among Estonian women [10], and they increased during 2008–2017 by 2.1% (95% CI 1.7, 2.4) among United States females [11]. Our analysis showed statistically significant continuous decline in overall and age-specific corpus uteri cancer mortality rates in Lithuania. There is scarce data on prevalence of the recognized risk factors [12] in Lithuania. However, the prevalence of one of the main risk factor for high corpus uteri cancer burden, overweight and obesity, is among the highest in Europe (59.6%) [13] and this may be related to the relatively high corpus uteri cancer mortality in Lithuania. Reduction in the body mass index, that was observed among younger women [14], may have contributed to a declining cohort effect in trend. According to our data, there was statistically significant period effect in corpus uteri cancer mortality rates. This implicates that in addition to falling incidence, improved diagnosis and treatment may have contributed to the declining mortality in Lithuania, although the improvement seems to have slowed in recent years. Most endometrial cancers (75%) are diagnosed at an early stage (FIGO stages I or II) [15]. In Lithuania, the proportion of new cases with stage I endometrial cancer increased from 44.6% in 2000 to 65.0% in 2012 [16,17]. It is likely that the more extensive use of pelvic ultrasonography, hysteroscopy, endometrial biopsy, sentinel lymph node biopsy, MRI, CT, and other approaches contributed to earlier detection or higher accuracy of diagnosis. Data from Estonia indicate that the increased proportion of surgically treated corpus uteri cancer have likely had a favorable impact on the increased survival rate during the period 1996–2002 (70%) to 2010–2016 (78%) [10]. In Lithuania, the age-adjusted 5-year relative survival for corpus uteri cancer increased statistically significantly by 8.7% between 1995–1999 and 2005–2009 [18]. The situation is comparable to that in Estonia, with survival estimate 73% in 2005–2007 [19]. However, Lithuania, like in other countries in Eastern Europe, substantially lower survival than most countries in Northern and Western Europe is detected, mainly due to low access or lack of latest diagnostic and treatment facilities [18,19]. Thus, to further improve survival and reduce mortality, ensuring prompt access to optimal diagnosis and treatment to all patients, continuous monitoring of corpus uteri cancer management and outcomes, as well as raising awareness of corpus uteri cancer symptoms, should be implemented.

We observed modest decrease in ovarian cancer mortality rates in Lithuania. Results are in line with those reported in previous studies, where decreasing rates were observed [2,11,20,21]. Significant period effect, detected in our study, indicates improvements in diagnosis or treatment of ovarian cancer. Most ovarian cancer cases are diagnosed at advanced stage, since the disease long remains asymptomatic and there are no means of diagnosing it at an early stage [19]. There were 12.1% stage I cancers among new cases in 2000 and 18.8% in 2012 in Lithuania [16,17]. Although survival remains poor compared to Northern and Central European countries, there was modest increase by 2.6% in five-year survival, from 33.2% in 1995–1999 to 35.8% in 2005–2009 [18,22]. Steadily decreasing mortality may be due to increasing survival that could be related to earlier detection, more adequate care, or both. The reduction in mortality may also be attributable to changes in already identified risk factors [2]. Excess body weight is a risk factor for ovarian cancer, thus a decrease in obesity among women in Lithuania [14], changes in diet and physical activity are potentially related to falling mortality. Similar to results in Poland [9], a decline in the ovarian cancer mortality in Lithuania was observed, despite the decreasing fertility rate and parity. Increased and earlier use of oral contraceptives possibly played certain role. There is evidence that among women aged 15–49 years, contraceptive (any) use was 59.6% in 2010 compared to 54.5% in 1990 [23]. An additional research is needed to better understand the causes of the ovarian cancer; nevertheless, based on available scientific data, a number of already identified risk factors could be altered to lower ovarian cancer mortality rates, e.g., further reduction in the body mass index among women, increasing parity, lactation, and regular physical activity, as well as improving cancer care and outcomes [2,19].

The study has several limitations. First, mortality data were used for this study because they are the only data that enable to evaluate most recent trends in the Lithuanian population as a whole. Second, this was an ecological descriptive analysis and further analytic epidemiological studies are needed to evaluate the effect of specific risk factors. Third, changes in mortality in the youngest and oldest birth cohorts should be interpreted with caution, as the values were based on few age-specific rates. Because of small number of cancer cases in youngest cohorts, and less reliable certification of the cause of death in elderly [24,25], the pattern observed may not be fully representative. Sharp changes for the youngest cohorts may be less stable; however, recent death rates in the young may carry important information for future trends. Fourth, changes to the ICD classification during the study period may have introduced a reporting bias. However, we did not find a significant difference in mortality rates after the ICD change. Also, the use of WHO data including deaths from cancers of uterus NOS may have resulted in overestimated corpus uteri mortality rates. However, it is plausible that in our study rates are reliable as the proportion of cancer of uterus NOS deaths for Lithuania is small. Data limitations support the importance of high quality cancer registry data in order to reduce the number of deaths coded as cancer of uterus NOS.

## 5. Conclusions

Cancers of the corpus uteri and ovary are among leading causes of cancer mortality in Lithuania and pose a serious epidemiological problem. Similar decreasing trends in corpus uteri and ovarian cancer mortality were observed suggesting shared risk factors and reduction in their prevalence or improvements in diagnosis and treatment. To further reduce the impact of the corpus uteri and ovarian cancers, recommendations to address preventable identified risk factors, such as obesity, diet, and lack of physical activity and further improvement in survival, including raising awareness of symptoms, ensuring prompt access to optimal diagnosis and treatment to all patients, and continuous monitoring of cancer management and outcomes, should be implemented.

## Figures and Tables

**Figure 1 medicina-56-00347-f001:**
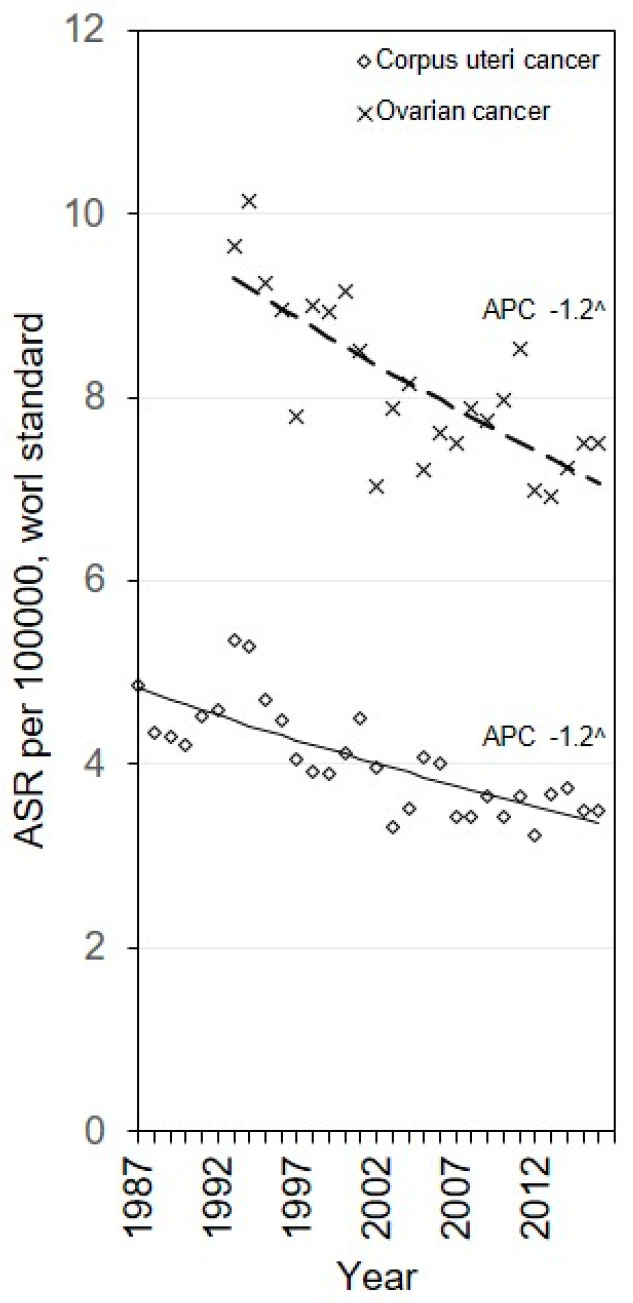
Modelled trends from Joinpoint regression (line) versus the observed age-standardized (ASR) mortality rates (dots) and annual percentage change (APC) in Lithuania, 1987–2016. ^ the APC is significantly different from zero.

**Figure 2 medicina-56-00347-f002:**
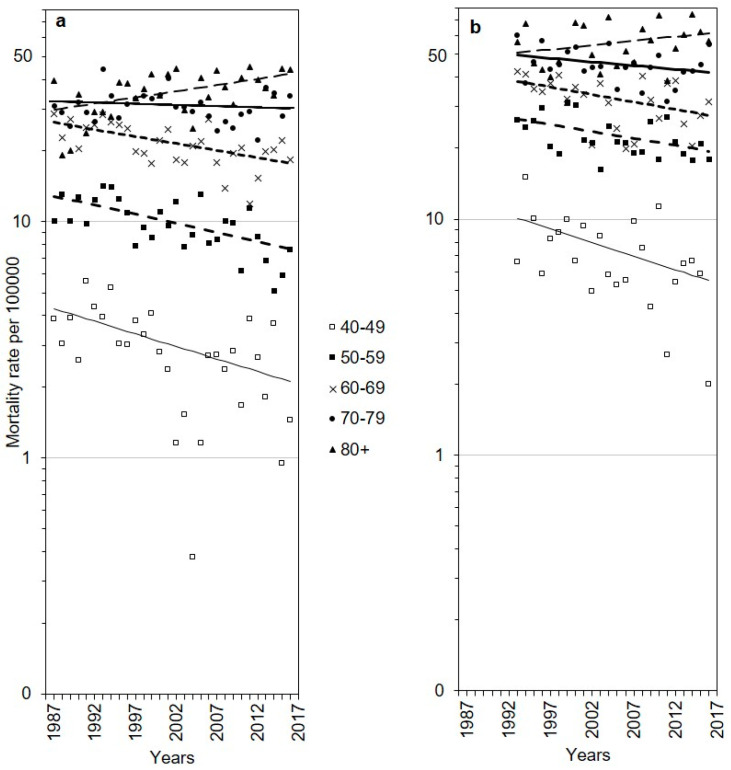
Age-specific observed (dots) and modelled from Joinpoint regression (lines) mortality rates in Lithuania, 1987–2016, plotted on a logarithmic scale: corpus uteri cancer (**a**); ovarian cancer (**b**).

**Figure 3 medicina-56-00347-f003:**
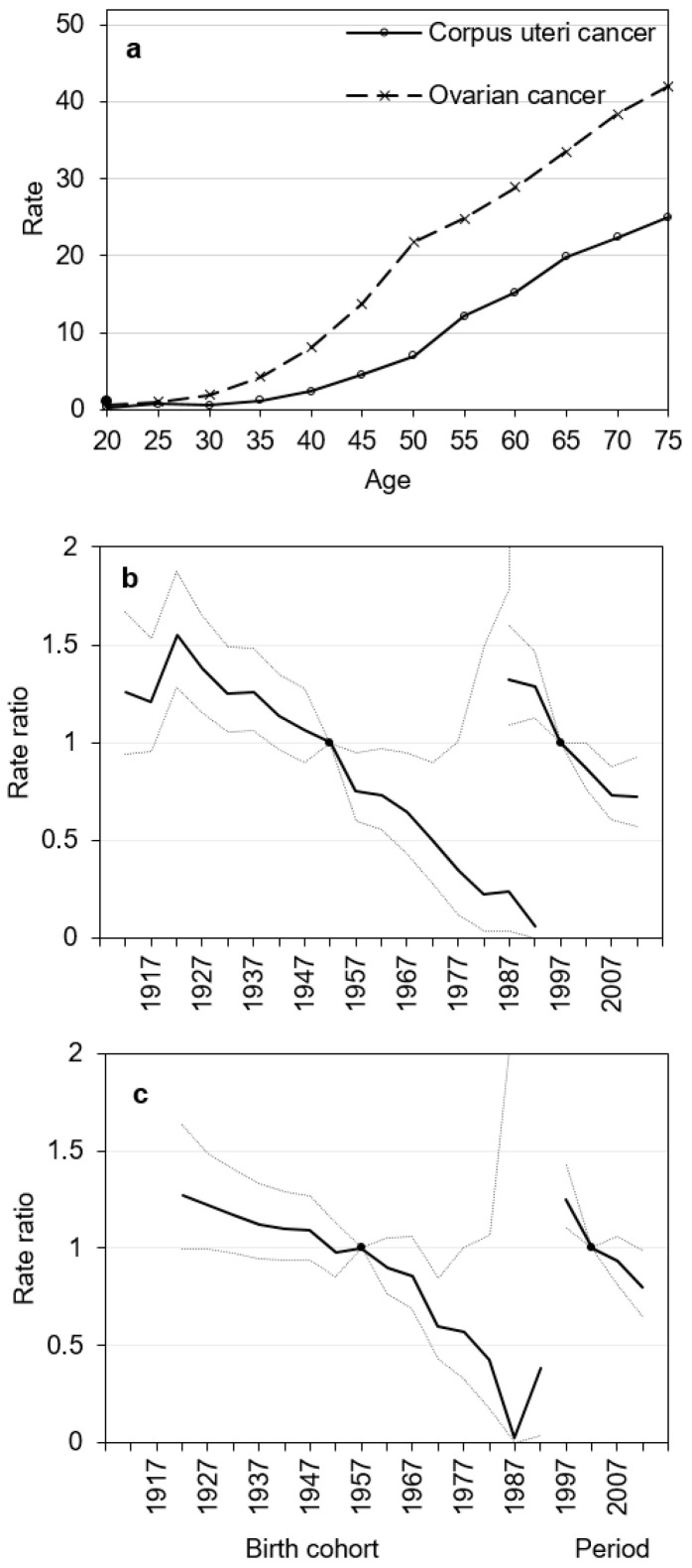
Longitudinal age curves (**a**) and estimated period and cohort effects and corresponding 95% confidence intervals from age–period–cohort analysis of mortality rates in Lithuania, 1987–2016: Corpus uteri cancer (**b**); ovarian cancer (**c**). On the *x*-axis, 5-year age groups, birth-cohorts and calendar periods are defined by the first year of the interval.

**Table 1 medicina-56-00347-t001:** List of the cancer sites included into analysis according to International Classification of Diseases (ICD) revision and years.

ICD Definition	ICD-10 Detailed 1998–2016	ICD-9 Basic 1993–1997	ICD-9 Special 1987–1992
Malignant neoplasm of cervix uteri	C53	B120	B120
Malignant neoplasm of uterus, other and unspecified	C54, C55	B122	B122
Malignant neoplasm of ovary and other uterine adnexa	C56	B123	B123

**Table 2 medicina-56-00347-t002:** Estimated annual percentage change (APC) in mortality rates from cancer of the corpus uteri and ovary in Lithuania, by age group and overall (ASMR).

Age at Death	Deaths	Rate		APC
	*n*	1987 ^a^	2016	From 1987 ^a^ to 2016
**Corpus uteri cancer**				
40–49	203	3.9	1.4	−2.4 (−4.0; −0.9)
50–59	661	10.1	7.6	−1.7 (−2.6; −0.9)
60–69	1288	28.7	18.2	−1.4 (−2.0; −0.7)
70–79	1367	30.8	34.1	−0.2 (−0.9; 0.4)
80+	833	39.3	44.0	1.2 (0.5; 2.0)
Total (all ages)	4405	4.9	3.5	−1.2 (−1.8; −0.7)
**Ovarian cancer**				
40–49	588	9.7	6.2	−1.5 (−3.0; 0.0)
50–59	1283	30.1	24.6	−1.0 (−1.8; −0.2)
60–69	1713	44.0	38.0	−1.2 (−2.0; −0.5)
70–79	1798	54.8	46.9	−0.7 (−1.2; −0.1)
80+	989	50.7	52.1	0.5 (−0.6; 1.6)
Total (all ages)	6527	9.7	7.5	−1.2 (−1.6; −0.8)

^a^ For Ovarian cancer: 1993.

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
