# Peer review of "Trends in Mortality Rates of Corpus Uteri and Ovarian Cancer in Lithuania, 1987–2016"

_medicina, 2020, doi:10.3390/medicina56070347_

Round 1

Reviewer 1 Report

Peer review of manuscript ID medicina-854356

Title: Trends in mortality rates of corpus uteri and ovarian cancer in Lithuania, 1987-2016

The authors investigated the trends in mortality rates of corpus uteri and ovarian cancer in Lithuania. This manuscript is well written and could show a similar decreasing pattern for endometrial and ovarian cancer mortality. 

Many thanks

Author Response

Reviewer: The authors investigated the trends in mortality rates of corpus uteri and ovarian cancer in Lithuania. This manuscript is well written and could show a similar decreasing pattern for endometrial and ovarian cancer mortality. 

Author's Reply: Thank you very much.

Reviewer 2 Report

Dear authors,

thank you very much for the opportunity to review your manuscript. The manuscripts regards a very methodological study to analyse the death rate of uterine and ovarian cancer.

I suggest to enforce the discussion, that is only based on the assumption of risk factors. I would rather discuss also about the advances in the treatment (think abouthow aggressive surgery can be in ovarian cancer, or what about SLN in endometrial cancetr) respectively for endometrial and ovarian cancer, and not only: also I would focus on early diagnosis, especially for endometrial cancer, given the awareness about symptoms.

Choort analysis performed in methods are not clearly exposed, please think to reshape the text.

In Figure 2 the legend is not complete (missing black triangle), further all the imaages are not self explanatory by reading the captions (especially Figure 3 is not clear and not commented as well in the text). I suggest to extensively comment the figures in the results and use them to support and guide your discussion.

Overall is a nice epidemiological paper, even though the Discussion should be largely empowered.

Best regards

Author Response

Reviewer: thank you very much for the opportunity to review your manuscript. The manuscripts regards a very methodological study to analyse the death rate of uterine and ovarian cancer.

Author's Reply: Thank you.

Reviewer: I suggest to enforce the discussion, that is only based on the assumption of risk factors. I would rather discuss also about the advances in the treatment (think abouthow aggressive surgery can be in ovarian cancer, or what about SLN in endometrial cancetr) respectively for endometrial and ovarian cancer, and not only: also I would focus on early diagnosis, especially for endometrial cancer, given the awareness about symptoms.

Author's Reply: We revised discussion carefully. Changes are indicated in red. We also included four more references.

Reviewer: Choort analysis performed in methods are not clearly exposed, please think to reshape the text.

Author's Reply: We revised the methods. We think that the age-period-cohort analysis method is used widely, so the very extensive description is not necessary.

Reviewer: In Figure 2 the legend is not complete (missing black triangle), further all the imaages are not self explanatory by reading the captions (especially Figure 3 is not clear and not commented as well in the text). I suggest to extensively comment the figures in the results and use them to support and guide your discussion.

Author's Reply: In Figure 2 the legend was corrected (black triangle included now). We also exteded comment of the figures in Results. Changes are indicated in red.

Reviewer: Overall is a nice epidemiological paper, even though the Discussion should be largely empowered.

Author's Reply: Thank you. We revised Discussion, focusing on cancer management. Changes are indicated in red.